# Theoretical and numerical considerations of rivers in a tectonically inactive foreland

**Stefan Hergarten**[1]

[1]Institut für Geo- und Umweltnaturwissenschaften, Albert-Ludwigs-Universität Freiburg, Albertstr. 23B, 79104 Freiburg, Germany

**Correspondence:** Stefan Hergarten
(stefan.hergarten@geologie.uni-freiburg.de)

**Abstract.** Modeling the dynamics of alluvial rivers is theoretically and numerically more challenging than modeling erosion of bedrock channels in active mountain ranges. As a consequence, the majority of the approaches developed in the context of alluvial rivers are one-dimensional. However, recent advances in the numerics of fluvial landform evolution models allow for two-dimensional simulations of erosion and sediment transport over time spans of several million years. This study aims at finding out fundamental properties of rivers in a tectonically inactive foreland of a mountain range by investigating a simple reference scenario theoretically and numerically. This scenario consists of a mountain range and a foreland in a quasi-steady state where the material eroded in the mountain range is routed through the foreland. In order to understand the properties of foreland rivers, a subdivision into two classes – carriers and redistributors – is introduced. Carriers originate in the mountain range and are thus responsible for the large-scale sediment transport to the ocean. In turn, redistributors are rivers whose entire catchment is located in the foreland. Using the concept of carriers and redistributors, it is shown that the drainage network in the foreland permanently reorganizes, so that a steady state in the strict sense is impossible. However, the longitudinal profiles of carriers are described well by a steady-state approximation. Their concavity index is considerably greater than that of rivers in the mountain range. Carriers are predominantly depositing sediment at high rates, while redistributors are eroding at much lower rates. Despite the low erosion rates, the sediment flux from redistributors into carriers is a major component of the overall sediment budget and finally the main driver of the highly dynamic behavior of the carriers.

## 1 Introduction

Fluvial deposits are among the most important records of Earth's tectonic and climatic history. Numerical models describing the physical processes controlling sediment production, transport, and deposition have become essential tools in this field. Finding out how perturbations in the depositional environment (e.g., changes in sea level) or in the source region (e.g., changes in precipitation or tectonic uplift) propagate through the system has been one of the most important applications of such models (e.g., Armitage et al., 2011, 2013; Mouchené et al., 2017; Yuan et al., 2022).

It seems, however, that our understanding of large-scale and long-term sediment deposition still lags behind our understanding of erosion processes in active mountain ranges. As reviewed by Romans et al. (2016) and by Tofelde et al.

(2021), large parts of our knowledge about the source-to-sink sediment transfer to the oceans are still on a conceptual level. However, analytical and numerical models have become increasingly important for understanding the general properties of alluvial rivers. Among the recently developed approaches, there are one-dimensional models focusing on individual rivers (e.g., Bolla Pittaluga et al., 2014; Blom et al., 2016, 2017; Malatesta et al., 2017; Wickert and Schildgen, 2019; Braun, 2022) as well as two-dimensional landform evolution models (e.g., Carretier et al., 2016, 2020; Yuan et al., 2019, 2022). In particular the latter may improve our understanding of sediment dynamics considerably in the near future. At the moment, however, little is known about the self-organization of alluvial rivers and its effects on rates of erosion and deposition. Even 40 years after Sadler (1981) described the decrease of rates of accumulation with increas-

ing time span of observation quantitatively, our understanding of this phenomenon is still incomplete.

Modeling sediment transport and deposition seems to be more challenging than modeling erosion. While all models reviewed by Coulthard (2001), Willgoose (2005), and van der Beek (2013) involve a sediment balance, it was already pointed out by Howard (1994) and by Kooi and Beaumont (1994) that simulating sediment transport in large rivers requires small time increments. This leads to a high computing effort and seriously limits the applicability of the models to large-scale problems with a reasonable spatial resolution. This problem is less severe in the limit of detachment-limited erosion (Howard, 1994) where it is assumed that all particles entrained by the river are immediately swept out of the system. Even a fully implicit scheme is available here, which in principle allows for arbitrarily large time increments and thus for large-scale simulations over long time spans (Hergarten and Neugebauer, 2001; Braun and Willett, 2013). Presumably owing to its theoretical and numerical simplicity, the concept of detachment-limited erosion has been applied in numerous studies of landform evolution. However, the central assumption that transported sediment has no effect on landform evolution limits the applicability of detachment-limited erosion to mountain streams.

Concerning the numerics of large-scale models including sediment transport, considerable progress was achieved recently. Yuan et al. (2019) combined the implicit scheme for erosion with a fixed-point iteration for the sediment fluxes. Their scheme achieves a high efficiency as long as the conditions are not too close to the transport-limited regime. The concept of transport-limited erosion assumes that the actual sediment flux of a river is always equal to the so-called transport capacity. This means that the rate of erosion or deposition instantaneously adjusts in such a way that the sediment flux equals the transport capacity. In contrast to the iterative scheme proposed by Yuan et al. (2019), the fully implicit scheme introduced by Hergarten (2020) even covers the entire range from detachment-limited to transport-limited erosion (and sediment deposition) at a constant numerical efficiency.

The recent numerical developments allow for large-scale simulations including sediment transport over long time spans and should thus also be able to improve our understanding of sedimentary systems. As a first result, Yuan et al. (2019) observed a permanent reorganization of the drainage pattern in a foreland region without uplift and subsidence over long times even under constant conditions. This reorganization causes an autocyclicity in erosion and deposition of sediments. From a theoretical point of view, the question arises whether such self-organizing systems are completely irregular or whether their average behavior can be described by simple relations or by one-dimensional models. The central question from a more practical point of view concerns the suitability of sediment archives for recording changes in climate and tectonics. In this context, not only the strength of

the autocyclic dynamics compared to allocyclic influences is relevant, but also the respective spatial and temporal scales.

The present study goes a step toward understanding the dynamics of a system that is completely dominated by autocyclic sediment dynamics. In the first part, it will be shown that steady-state topographies in the sense of constant elevation at each point are impossible in absence of uplift and subsidence, so that there must be autocyclic aggradation and incision. In the next step, properties of hypothetical steady-state river profiles will be investigated. As a main finding, such rivers capture the properties of large rivers in a regime of permanent reorganization quite well on average, although they cannot be stable over long times. The second part of the paper provides preliminary estimates for rates of erosion and deposition and for the time scale of network reorganization.

## 2 Model setup

This study addresses the simplest scenario of rivers in a tectonically inactive foreland. A rectangular domain with a north-south oriented mountain range at the center and a foreland at each side is considered (Fig. 1). While the mountain range is uniformly uplifted at a constant rate, the foreland regions are neither uplifted nor subsiding. The northern and southern boundaries are periodic, while the eastern and western boundaries are kept at zero elevation and are interpreted as the coast of an ocean.

The open-source landform evolution model OpenLEM (Hergarten, 2022a) is employed for all numerical simulations. Since focus is on a minimum scenario, none of the components of OpenLEM beyond fluvial erosion and sediment transport, such as lithospheric flexure and orographic precipitation (Hergarten and Robl, 2022), are used. The fluvial model implemented in OpenLEM is presumably the simplest model of large-scale fluvial erosion and sediment transport. Several formulations of this model were proposed, which are all similar in their spirit or partly even mathematically equivalent: the undercapacity model (Kooi and Beaumont, 1994), the linear decline model (Whipple and Tucker, 2002), the $\xi$–$q$ model (Davy and Lague, 2009), and the shared stream-power model (Hergarten, 2020). In this study, the shared stream-power formulation

$$\frac{E}{K_{\mathrm{d}}} + \frac{Q}{K_{\mathrm{t}}A} = A^m S^n \tag{1}$$

is used, where $E$ is the erosion rate, $Q$ the sediment flux (volume per time), $A$ the upstream catchment size, and $S$ the channel slope. The model involves four parameters $K_{\mathrm{d}}$, $K_{\mathrm{t}}$, $m$, and $n$. The term $A^m S^n$ is often called stream-power term, and the model implements the idea that this term is used jointly by erosion and sediment transport.

While the equation for the change in surface elevation $H$ at a given uplift rate $U$ is straightforward,

$$\frac{\partial H}{\partial t} = U - E, \tag{2}$$

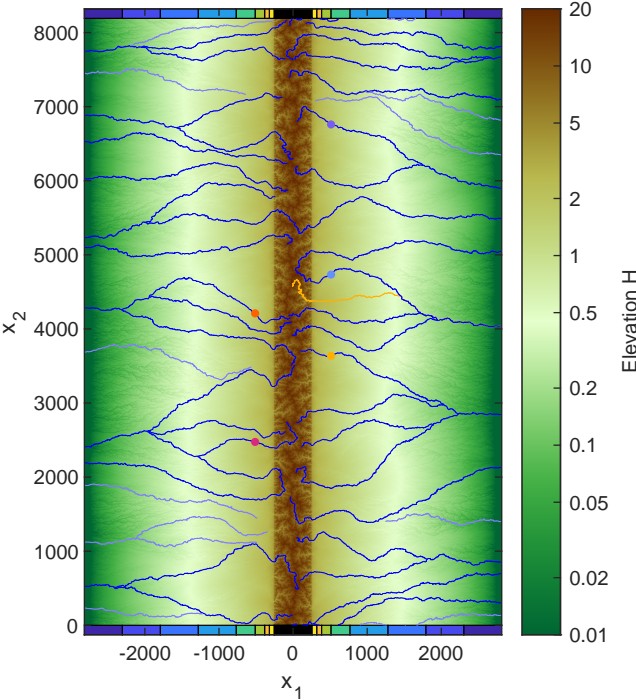

**Figure 1.** Snapshot of the topography at $t = 128$ including the 50 largest rivers. Solid blue lines refer to carriers and pale blue lines to redistributors according to the distinction made in Sect. 4. The dots mark the five biggest rivers, which are analyzed in more detail in Fig. 7. The orange-colored river is the largest river with regard to the catchment size at the edge of the mountain range (Fig. 4). The additional colorbars at the top and bottom define regions of different distances from the mountain range and are used in several other figures.

an additional balance equation for the sediment fluxes is required in order to obtain a closed system of equations. Assuming that each node $i$ of a discrete grid delivers its entire sediment flux $Q_i$ to a single neighbor, this sediment balance equation reads

$$E_i = \frac{Q_i - \sum_j Q_j}{s_i}, \tag{3}$$

where $s_i$ is the size (area) of the respective grid cell. The right-hand side of Eq. (3) is a discrete representation of the divergence operator, where the sum extends over all neighbors that deliver their sediment flux to the cell $i$. This sum is empty for nodes that do not receive sediment, which may be seen as internal boundaries. For consistence, the same routing scheme should be used for water (catchment size $A$ and direction of channel slope $S$) and sediment flux. The implementation in OpenLEM uses the D8 routing scheme (O'Callaghan and Mark, 1984) on a regular grid.

In absence of transported sediment ($Q = 0$), the shared stream-power model reduces to

$$E = K_d A^m S^n, \tag{4}$$

which is the stream-power incision model widely used in the context of detachment-limited erosion. The parameter $K_d$ is called erodibility, where the subscript emphasizes the relation to detachment-limited erosion. In turn, neither erosion nor deposition takes place ($E = 0$) if the sediment flux is

$$Q = K_t A^{m+1} S^n. \tag{5}$$

So this term defines the transport capacity, where the transport coefficient $K_t$ describes the ability to transport sediment at given $A$ and $S$. The shared stream-power model turns into the stream-power incision model for $K_t \to \infty$ and into a transport-limited model for $K_d \to \infty$. In the latter limit, the sediment flux is always equal to the transport capacity defined by Eq. (5). As discussed by Hergarten (2020), the system of equations reduces to a single diffusion equation then.

For spatially uniform erosion, the sediment flux is $Q = EA$, and Eq. (1) collapses to a form analogous to the stream-power incision model (Eq. 4) with an effective erodibility $K$ according to

$$\frac{1}{K} = \frac{1}{K_d} + \frac{1}{K_t}. \tag{6}$$

River profiles follow the relation

$$S \propto A^{-\theta} \tag{7}$$

with $\theta = \frac{m}{n}$ then. The exponent $\theta$ in Eq. (7) is called concavity index in the context of analyzing river profiles. Concavity indexes of real rivers have been investigated in numerous studies, starting from the seminal work of Hack (1957). Values $\theta \approx 0.5$ are typically found for rivers at uniform erosion, where often either $\theta = 0.45$ or $\theta = 0.5$ is used as a reference value (e.g., Whipple et al., 2013; Lague, 2014). So the ratio of the exponents $m$ and $n$ is constrained quite well by the concavity of real-world rivers.

The absolute values of the exponents $m$ and $n$ are, however, more uncertain than their ratio since they cannot be determined from the shape of river profiles under uniform erosion. Assuming $n = 1$ simplifies both theoretical considerations and the numerical implementation since the model is linear with regard to the topography then. In turn, the results compiled by Lague (2014) as well as some recent studies (Harel et al., 2016; Hilley et al., 2019; Adams et al., 2020) rather suggest $n > 1$.

In this study, the linear version of the shared stream-power model ($n = 1$) is used for numerical reasons. The fully implicit scheme introduced by Hergarten (2020) is not only stable at arbitrary time increments, but also avoids oscillations in elevation or rates of erosion and deposition, e.g., if a river is suddenly exposed to a large sediment flux after an avulsion event. It is, however, restricted to $n = 1$. The implementation in OpenLEM contains a semi-implicit extension for $n > 1$ which is still stable at large time increments, but not able to avoid oscillations completely. While these oscillations are

not a problem in many applications, they affect the short-term rates of erosion investigated in Sect. 8 and may even cause an artificially increased frequency of avulsion events. While some simulations were also performed for $n = 2$, the results are not included in this paper.

Since the foreland region will be covered by alluvial deposits, different lithologies are assumed in the two domains, in contrast to the recent study of Yuan et al. (2022). As the simplest approach, it is assumed that $K_t$ is the same in both domains, while the values of $K_d$ differ. The ratio of $K_d$ and $K_t$ is mathematically equivalent to the parameter $\Theta$ or $G$ (depending on the notation) in the $\xi$–$q$ model. The results obtained by Davy and Lague (2009) and Guerit et al. (2019) suggest $K_d \gtrapprox K_t$ for $n = 1$. For simplicity, $K_d = K_t$ is assumed in the mountain range, which was assumed by Yuan et al. (2022) for the entire domain.

We will see in Sect. 3 that the choice $K_d = K_t$ for the mountain range has a minor effect on the rivers in the foreland. In turn, the parameter choice for the foreland is more critical. As discussed by Hergarten (2021), $K_d$ refers to the properties of the actual river bed. As a consequence, $K_d$ should be much larger than $K_d$ of the bedrock if a previously deposited thick alluvial cover is eroded. In an environment of deposition ($E < 0$), Eq. (1) should even be replaced by the transport-limited version ($K_d \rightarrow \infty$). Otherwise, assuming a lower erodibility $K_d$ would reduce the rate of deposition if all other parameters remain constant, which would not make much sense. In order to keep the model as simple as possible, the transport-limited end-member ($K_d \rightarrow \infty$) is used in the entire foreland region, no matter whether the rivers are actually depositing sediments or eroding. An alternative scenario will be considered in Sect. 10.

The implicit scheme inhibits the formation of lakes in individual streams. However, lakes may occur at confluences. If a large amount of sediment is deposited in a large river, its elevation may exceed that of its tributaries. Then the lowermost ranges of the tributaries turn into lakes. Formally, this leads to an upstream sediment flux, so that the lowermost ranges of the tributaries serve as accommodation area for the sediments. Since this phenomenon is not unrealistic and does not affect the stability of the scheme, there is no need to inhibit these backward sediment fluxes. It should, however, be noted that this effect does not add floodplains to the model in a realistic way since it is only based on the topography of the bed and does not take into account the water level.

In order not to introduce unnecessary constraints, a nondimensional representation is used in all simulations and theoretical considerations. Assuming $n = 1$, the choice $m = 0.5$ (so $\theta = 0.5$) is convenient since it avoids odd physical units of $K_d$ and $K_t$ and simplifies the scaling from nondimensional to real-world properties. For this choice of $m$ and $n$, the dimensions of $K_d$, $K_t$, and $K$ (Eq. 6) are inverse time, so that each of them can be used for defining the time scale independently of the spatial scales. Since $K_t$ is constant in the entire domain, the nondimensional representation is based on

$K_t = 1$. Then one nondimensional time unit corresponds to an absolute time span of $T = \frac{1}{K_t}$ (with the real-world value of $K_t$). However, real-world estimates are typically obtained from the steepness of rivers at a given erosion rate and thus refer to $K$ (Eq. 6) instead of $K_d$ or $K_t$. Assuming $K_d = K_t$ leads to

$$T = \frac{1}{K_t} = \frac{1}{2K}. \tag{8}$$

With the value $K = 2.5 \ \mathrm{Myr}^{-1}$ used by Robl et al. (2017), this would yield a time scale $T = 200,000$ yr. So some 100,000 years should be a realistic magnitude for one nondimensional time unit.

The vertical length scale $L_v$ is defined by the uplift rate $U$ and the time scale $T$ in the form

$$L_v = UT. \tag{9}$$

Using a nondimensional uplift rate $U = 1$, a convenient vertical length scale of $L_v = 100$ m would emerge at a real uplift rate of 0.5 millimeters per year for $T = 200,000$ yr.

The horizontal length scale is arbitrary and independent of all model parameters and of the other scales for $\frac{m}{n} = 0.5$. In principle, this allows for an arbitrary scaling of the horizontal coordinates. However, the pixel size of the model imposes two limitations. First, hillslope processes are not taken into account, so that the shared stream-power is assumed to hold even for single-pixel catchments. Therefore, the pixel size should not be too small. More important for this study, the pixel size also defines the area over which deposited sediments are distributed since the model does not include floodplains explicitly.

All simulations were performed on a grid with 8192 rows and 5632 columns (Fig. 1), starting from a flat topography superimposed by a small random disturbance with a range of $10^{-4}$. The mountain range has a width of 512 cells. So each side of the foreland is five times as wide as the mountain range. This choice ensures that a large part of the foreland consists of an alluvial plain that is not affected strongly by the properties of the mountain range and its alluvial fans. In turn, the width of the mountain range is large enough to generate rivers with a wide spectrum of catchment sizes in the mountain range (up to about 100,000 pixels).

The choice of the time increment $\delta t$ is not trivial, although the fully implicit scheme implemented in OpenLEM is unconditionally stable for any $\delta t$. As discussed by Hergarten (2020), the limitation arises from changes in flow direction rather than from the accuracy of the scheme itself. Since each cell can change its flow direction only once per timestep, large time increments slow down the dynamics of the system. Preliminary tests with a similar setup revealed that this effect becomes dominant for $\delta \gtrapprox 10^{-2}$. In order to stay clearly below this limit, $\delta t = 2^{-10} \approx 10^{-3}$ is used in this study, although the results are by far not independent of $\delta t$ then (see also Sects. 3 and 9). According to the time scale estimated

above, $\delta t$ is in an order of magnitude of some hundred years. Using much smaller values of $\delta t$ would not only increase the numerical effort, but would also question the applicability of the model since we would probably have to take into account
the effects of individual flood events on river avulsions. However, simulating individual events is outside the scope of the stream-power concept.

## 3 First results

A strong reorganization of the drainage pattern occurs during
the first phase of the simulation. This reorganization slows down in the tectonically active region with increasing incision of the rivers and has almost ceased at $t \approx 10$. Afterwards, there is little reorganization in the mountain range, although the topography is still far off from equilibrium. Only
32 changes in flow direction occur at points with a catchment size between 1000 and 10,000 pixels (where the largest catchment in the mountain range is about 100,000 pixels large) from $t = 10$ to the end of the simulation at $t = 500$, so less than 1 change per 15 time units.
In turn, the reorganization of the drainage network continues in the foreland. On average, 2.5 % of all sites change their flow direction in each timestep, while avulsions of large rivers take place more frequently. Even 24 % of all sites with $A \geq 100,000$ change their flow direction in each timestep on
average. However, the respective frequencies of avulsion depend on the time increment $\delta t$. Performing a shorter simulation with an eight times smaller $\delta t = 2^{-13}$ yielded percentages of 0.51 % and 9.9 %, respectively. So the mean rate of avulsions taken over all sites increases moderately by a
factor of 1.6 if $\delta t$ is reduced by a factor of 8. For the large rivers, however, it is a factor of about 3. However, we will see in Sect. 9 that the effect of $\delta t$ on the long-term reorganization of the drainage pattern is weaker than it seems here. Nevertheless, we should keep in mind that in particular the
frequency of avulsions of large rivers depends on the time increment and that the frequency of large floods may play an important part in nature.
  The highest mean and peak elevations are reached in the mountain range at $t \approx 80$. Afterwards, the topography is
40 still not constant, but decreases very slowly due to the slow network reorganization. This effect was described for the stream-power incision model by Robl et al. (2017), but here it is not relevant since it is much slower than the ongoing network reorganization in the foreland. Finally, the time span
from $t = 100$ to $t = 500$ was used in the following analyses, where most of the data were obtained by averaging over 401 equally spaced snapshots ($\Delta t = 1$).
  Figure 2 shows a swath profile of the mean topography over the considered time span. Minimum and maximum val-
50 ues are not taken over $x_2$ and $t$, but only over $x_2$ and then averaged over all snapshots. The foreland topography becomes increasingly steep close to the mountain range and reaches a

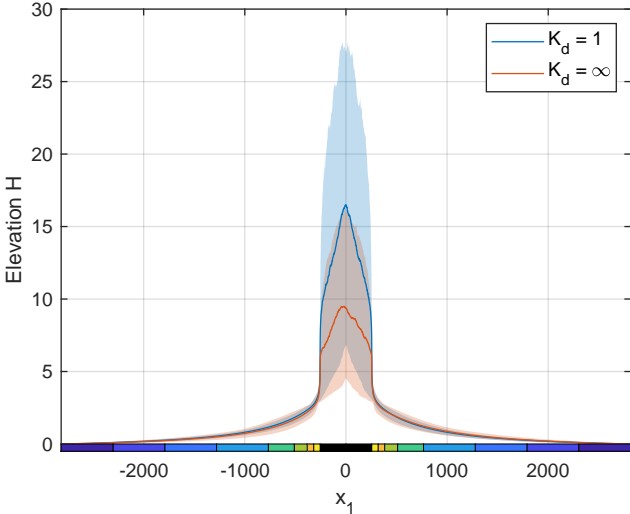

**Figure 2.** Mean swath profiles of the topography obtained by averaging over the 401 snapshots. Solid lines show the mean elevation and filled areas the range from minimum to maximum elevation.

mean value of about 5.6, corresponding to about one-fifth of the peak elevation and one-third of the mean elevation along the center of the mountain range.
  The topography obtained under transport-limited conditions in the mountain range ($K_t = 1$ and $K_d = \infty$ as in the foreland) is also shown in Fig. 2 for comparison (however, based on only 160 snapshots). This change mainly affects the topography in the mountain range. While the effective erodi-
60 bility was $K = 0.5$ according to Eq. (6) before, it is twice as high ($K = 1$) for transport-limited conditions. As a consequence, equilibrium channel slopes and thus relief are two times lower. As discussed by Hergarten (2021), $K_d$ determines the speed at which knickpoints migrate upstream and
65 thus the response of the sediment flux from the mountain range to changes in topography in the foreland. However, the feedback on the topography in the foreland appears to be minor. In this context, we should keep in mind that assuming transport-limited erosion in the mountain region is an unre-
70 alistic extreme scenario, although the ratio of $K_d$ and $K_t$ is still not constrained well from real-world data. So these results suggest that a moderate deviation from $K_d = K_t$ will have an effect on the height of the mountain range compared to the topography of the foreland, but will not affect the prop-
75 erties of the foreland seriously.
  Figure 3 provides a more detailed analysis of the mean topography (averaged over both sides). The relief (maximum minus minimum elevation taken in the $x_2$-direction) is greater than the mean elevation in the mountain range, corre-
80 sponding to deeply incised valleys and high peaks. This relation is inverted in the foreland, corresponding to the rapid decline in maximum elevation visible in Fig. 2. In the orange-colored region (see Fig. 1), relief is even less than one-third of the mean elevation. So the topography is quite smooth

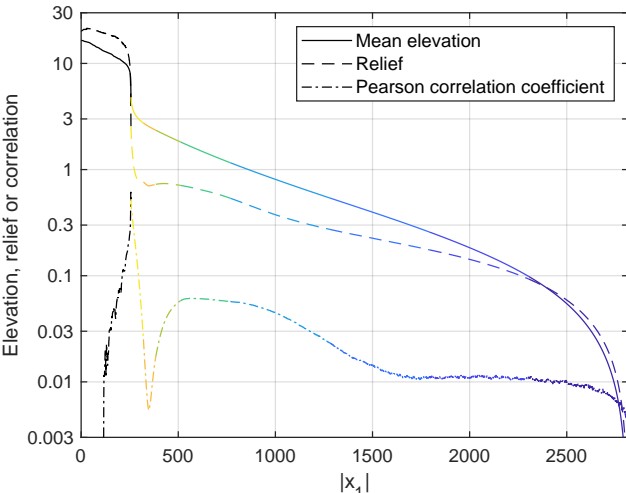

**Figure 3.** Mean elevation, relief (maximum minus minimum elevation taken in the $x_2$-direction), and Pearson correlation coefficient of the elevation at the considered position $x_1$ and the topography at the edge of the mountain range ($|x_1| = 256$). The colors correspond to the regions defined in Fig. 1.

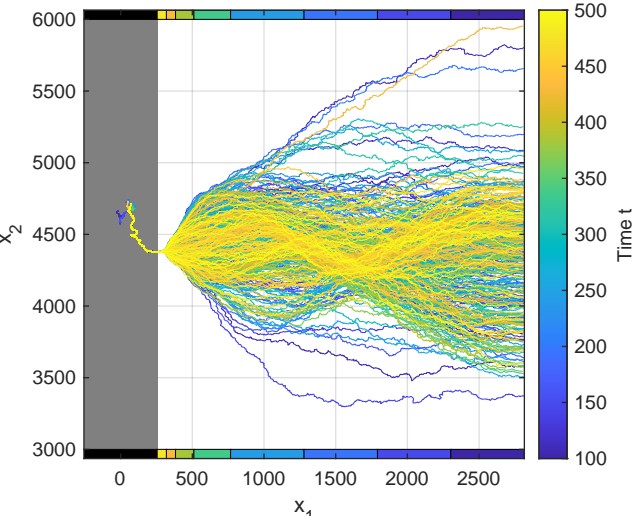

**Figure 4.** Snapshots of the river with the largest catchment at the edge of the mountain range (orange-colored line in Fig. 1). The gray-shaded area depicts the mountain range.

here. This decrease in relief goes along with a distinct minimum in the correlation of the topography to the topography at the edge of the mountain range ($|x_1| = 256$). Surprisingly, the correlation slightly recovers in the green and turquoise domains, and a weak positive correlation persists in the entire domain.

Further away from the mountain range, both mean elevation and relief decrease toward the ocean. The decrease in relief is, however, slower than the decrease in mean elevation. So the topography becomes smoother toward the ocean on an absolute scale, but rougher in relation to the mean elevation.

Figure 4 gives a first insight into the dynamics of the rivers in the foreland. The river considered here is not the biggest river overall (which is not the same at all times), but the river with the largest catchment at the edge of the foreland (orange-colored line in Fig. 1). Since network reorganization in the mountain range is weak, this river remains the same for all times $t \geq 100$ considered here, where changes take place only in the uppermost part of its catchment.

After leaving the mountain range, the area covered by the river widens moderately. As discussed above, the rivers are confined in narrow gorges in the mountain range, but their relief rapidly decreases in the foreland, allowing for wider valleys. At some point, however, the behavior changes abruptly to form a large alluvial fan. Directions of flow vary by more than $90°$ here, while the river itself is more or less straight.

In contrast to typical alluvial fans in real-world topographies, the alluvial fan observed here is not sharply bounded downstream, but rather dissolves by diverting the river systematically toward the ocean. This difference is related to the topography already being in a quasi-steady state, where the fan is no longer growing. Analyzing the predicted sizes of

the alluvial fans systematically and comparing them to real-world data would be interesting. The data reviewed by Blair and McPherson (2009) suggest a linear relation between the areas of the fans and the respective upstream catchment sizes. Braun (2022) reproduced such a linear relation by assuming a specific relation between catchment size and river length in the foreland. In turn, the behavior of the numerical model considered here suggests that the distance of the river to the nearest river of similar size may be the primary control rather than the absolute catchment size. While the results may finally be similar, a thorough analysis could be a starting point of a subsequent study.

## 4   The concept of carriers and redistributors

In our scenario of an active mountain range and entirely passive foreland regions, it makes sense to classify rivers into two categories. Let us define carriers as those rivers that receive discharge (and thus sediment flux) from the mountain range. In turn, rivers whose entire catchment is located in the passive foreland are called redistributors in the following.

The majority of the large rivers are carriers. In Fig. 1, 39 out of the 50 biggest rivers are carriers. However, it is visible that the sources of large redistributors are either close to carriers or to valleys in the mountain range. This observation suggests that large redistributors were either carriers in the past and were disconnected by avulsion events or will turn into carriers in the future.

Carriers and redistributors differ fundamentally in their properties since carriers do not only receive discharge from the mountain range, but also a sediment flux. In turn, the sediment flux of redistributors arises solely from erosion in the

tectonically inactive foreland. Since the foreland topography is typically not very steep, erosion rates and sediment fluxes of redistributors are rather low. As long as the base level of a redistributor (either at the ocean or at the confluence with a carrier) remains constant, the topography of its catchment is decaying. So redistributors are predominantly erosive with moderate rates.

Conversely, carriers must deposit sediment on average since the erosion of the redistributors would result in an ongoing decrease in topography in the foreland otherwise. Furthermore, carriers must be steeper than redistributors on average, owing to their higher sediment flux. This difference is responsible for the more elongated shapes of catchments in the foreland compared to the mountain range, which is immediately recognized in Fig. 1. In the mountain range, large rivers are less steep than small rivers, so that small rivers tend to flow into large rivers instead of flowing directly toward the edge of the mountain range. In contrast, the largest rivers in the foreland are carriers. Since these are rather steep, the tendency of redistributors to flow toward large carriers instead of draining into the ocean is much weaker than in the mountain range, which results in strongly elongated catchments.

## 5 The impossibility of steady-state solutions

Real-world topographies are typically not in a steady state. The event- or threshold-based characteristics of at least some involved processes is a primary reason for the absence of steady states. As an example, an individual big flood may contribute much to landform evolution. As a second aspect, approaching a steady state may take a long time, while the tectonic and climatic conditions are typically not constant over sufficiently long time spans.

Using the $\xi$–$q$ model, Yuan et al. (2019) already observed a permanent reorganization of the rivers in a passive foreland, although all conditions (including the uplift rate in the mountain range) were constant. In contrast, constant uplift rates typically result in steady-state topographies. So there seems to be a fundamental difference in the properties of the model between active and passive regions. This difference is not only relevant for our understanding of landform evolution, but also for the question whether a record that suggests non-steady conditions is necessarily related to changes in tectonic or climatic conditions or whether it may be the result of self-organization.

Using the properties of carriers and redistributors described in the previous section, it can easily be shown that steady-state topographies are indeed impossible in a passive foreland for models of the type considered here. If the topography was in a steady state, the topography of all redistributors would be flat. This would imply that the catchments of all redistributors that drain directly into the ocean would be at sea level, while those that drain into a carrier would be exactly at the elevation of the point of confluence. This situation

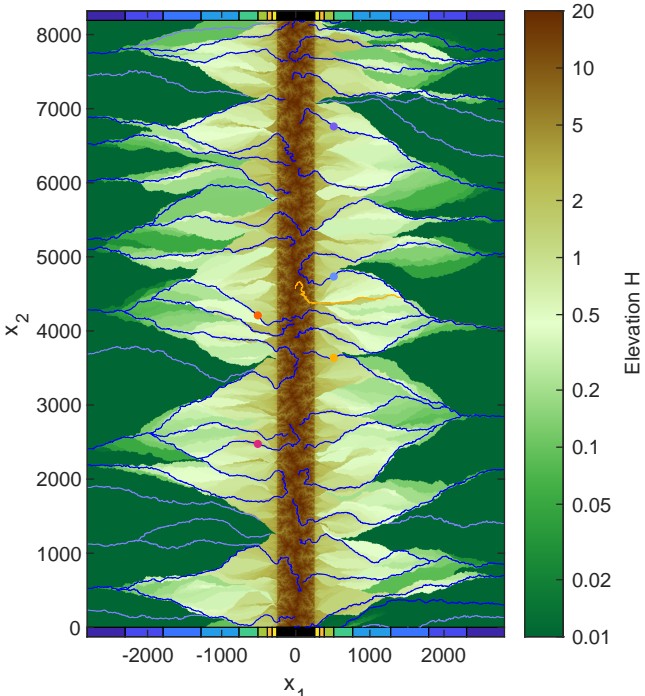

**Figure 5.** Steady-state topography based on the flow pattern of the topography shown in Fig. 1. All markers are the same as in Fig. 1 and were just included for completeness.

is illustrated in Fig. 5 for the drainage pattern from Fig. 1. It would immediately result in inconsistent flow directions at drainage divides. This inconsistence is obvious at the borders of the catchments of redistributors that drain directly into the ocean (dark green areas). These drainage divides would move into the adjacent higher-elevation catchments. However, even the catchments of redistributors draining into the same carrier would be inconsistent since the catchments that drain into the carrier more downstream are lower in elevation. So there would be a shift in the drainage divides, where the redistributors play an essential part. These arguments are not restricted to the specific model considered here, but only rely on the property that the catchments of redistributors become flat through time.

## 6 Properties of carriers in a steady state

In the previous section, we have seen that the drainage network and thus also the topography cannot reach a steady state in the foreland even under constant conditions. Nevertheless, it is useful to investigate the properties of the rivers in a hypothetical steady state in order to find out whether they have anything in common with the rivers in a state of permanent reorganization. So let us assume that the drainage network was frozen and consider the properties of the carriers for a steady-state topography like the one shown in Fig. 5, al-

though the flow directions are not consistent with the topography at the drainage divides.

According to Eq. (5), the channel slope in a steady state at zero uplift ($E = 0$) follows the relation

$$S^n = \frac{Q}{K_t A^{m+1}}. \tag{10}$$

If $Q$ was constant along the river (the sediment flux from the mountain range), the concavity of the river profile would arise from the downstream increase in discharge alone according to

$$S \propto A^{-\theta} \tag{11}$$

with $\theta = \frac{m+1}{n}$. Then the concavity index $\theta$ would be by $\frac{1}{n}$ greater than the concavity index at uniform erosion ($\theta = \frac{m}{n}$). This would, however, only be true if all tributaries were redistributors, which would not contribute sediment in a steady state. Confluences with other carriers lead to a downstream increase in sediment flux and thus to a weaker concavity.

Let us consider a cross section in the $x_2$-direction (parallel to the mountain range). If $d$ is the mean spacing of the carriers crossing this line, each carrier has to accommodate a sediment flux of

$$Q = U \frac{w}{2} d \tag{12}$$

on average, where $w$ is the width of the mountain range (so $\frac{w}{2} d$ is half of the area of the mountain range) and $U$ the uplift rate.

In order to relate the equilibrium sediment flux $Q$ to the catchment size $A$, the relation between $d$ and $A$ was investigated numerically. For this purpose, the number of carriers and their mean catchment size were measured for each line of the grid over the 401 snapshots. Note that the mean catchment size cannot be estimated from the total area upstream of the line and the respective number of carriers alone since a part of the domain is directly drained into the ocean by redistributors.

The results shown in Fig. 6 suggest that there are two regimes with simple scaling relations between $d$ and $A$. A linear relation $d \propto A$ is found close to the mountain range. More important, the relation turns into a power law $d \propto A^\gamma$ with $\gamma = 0.53$ at greater distances, starting from about half the width of the mountain range, so for mean catchment sizes $A \gtrsim 50{,}000$. The question whether it is a fractal relation with $\gamma > 0.5$ or $\gamma = 0.5$ ($d \propto \sqrt{A}$) is not important here.

Inserting this result into Eq. (12) yields

$$Q \propto A^\gamma. \tag{13}$$

This relation can be reconciled with the analysis of real-world drainage networks by Prasicek et al. (2020), although that study refers to mountain catchments. It was found there that confluences of rivers of the same stream order (so of

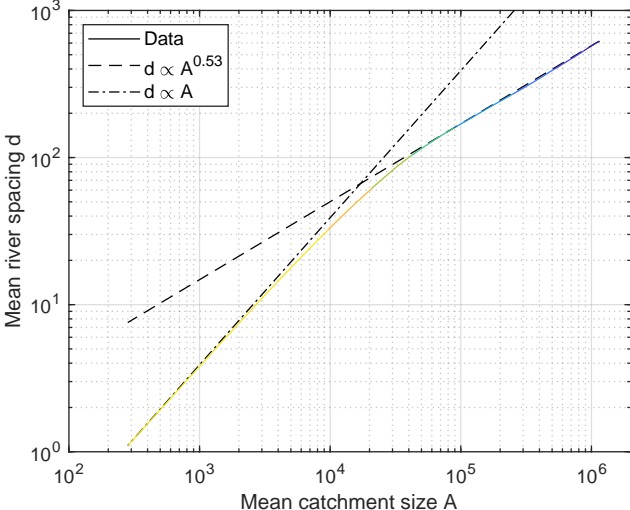

**Figure 6.** Relation between the mean spacing $d$ of the carriers and the mean catchment size $A$. Averaging was performed along lines in the $x_2$-direction over 401 snapshots of the topography.

similar sizes) and capture of smaller tributaries contribute almost equally to the downstream increase in catchment size. Since the largest rivers are typically carriers, while redistributors are smaller, this means that confluence with other carriers and capture of redistributors contribute almost equally to the downstream increase in catchment size. As redistributors do not deliver sediments in a hypothetical steady state, this implies

$$\frac{\delta Q}{Q} \approx \frac{1}{2} \frac{\delta A}{A} \tag{14}$$

for the downstream increases $\delta Q$ and $\delta A$, which is equivalent to Eq. (13) for $\gamma \approx 0.5$. This agreement with the findings of Prasicek et al. (2020) suggests that Eq. (13) is not specific to the linear version ($n = 1$) of the model used here, but reflects a general property of drainage networks. So Eq. (13) may also be helpful for developing one-dimensional versions of more complex models.

It should, however, be kept in mind that water and sediments are routed toward a single neighbor in the model used here. So Eq. (13) and the following considerations are only valid for dendritic channel networks. Finding out whether bifurcations in the network only affect the factor of proportionality or also the exponent $\gamma$ and thus also the concavity index would require further research.

Inserting Eq. (13) into Eq. (10) yields

$$S^n \propto A^{\gamma - m - 1}. \tag{15}$$

So the fundamental relation for the concavity of rivers (Eq. 11) also holds for carriers in a steady state, but with a concavity index

$$\theta = \frac{m + 1 - \gamma}{n} \approx \frac{m}{n} + \frac{1}{2n}. \tag{16}$$

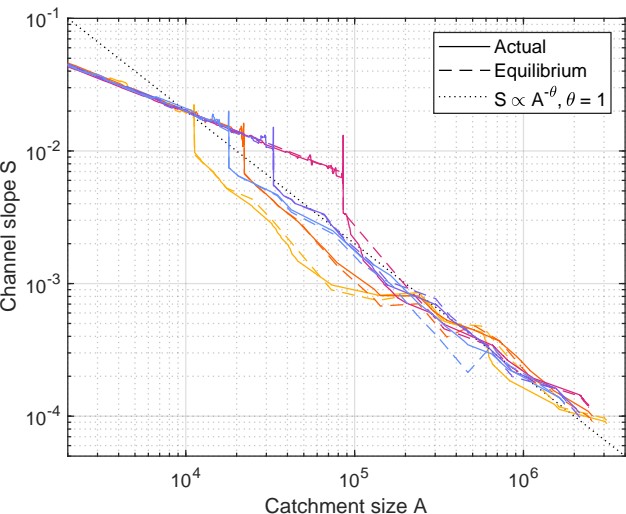

**Figure 7.** Channel slopes of the five biggest rivers shown in Fig. 1 ($t = 128$, solid lines) and Fig. 5 (steady state, dashed lines).

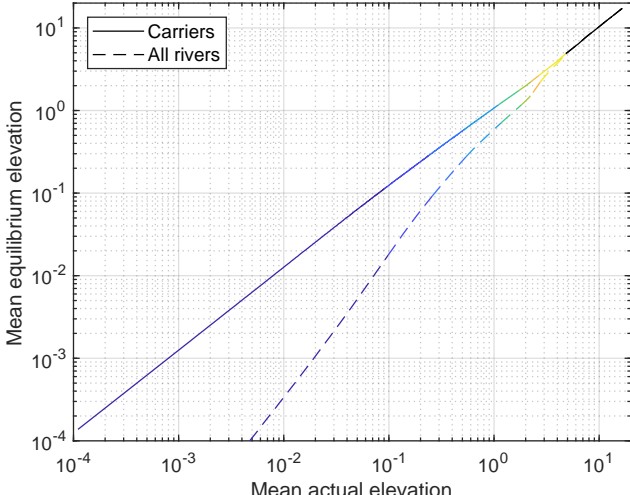

**Figure 8.** Mean equilibrium elevation vs. mean actual elevation for carriers (solid lines) and all rivers (dashed lines). The data were obtained from averaging over lines in the $x_2$-direction and over 401 snapshots. The colors refer to the regions defined in Fig. 1.

So the concavity index of carriers in a steady state is by about $\frac{1}{2n}$ higher than at uniform erosion. For the parameter values considered here ($m = 0.5$, $n = 1$), the concavity index is $\theta \approx 1$.

Since it was shown in the previous section that steady states cannot exist in the foreland, the question arises whether the properties of steady-state carriers obtained above are relevant at all. Figure 7 shows an example of the five biggest rivers from Fig. 1 ($t = 128$, solid lines) and Fig. 5 (steady state, dashed lines).

Since the uppermost parts of the rivers are located in the mountain range, they follow straight lines with a negative slope of $\theta = \frac{m}{n} = 0.5$ in the double-logarithmic plot in equilibrium (dashed lines). The respective curves from the snapshot (solid lines) show the same overall behavior in the mountain range, but with distinct local deviations in channel slope. These are disturbances propagating upstream, so mobile knickpoints. While these mobile knickpoints originate from changes in foreland topography, their feedback on the rivers in the foreland by means of changes in sediment flux is small.

In the foreland region, steady-state river segments with a concavity index $\theta = \frac{m+1}{n} = 1.5$ are found, as predicted for carriers that do not receive sediment flux from their tributaries. These segments are displaced horizontally and vertically at confluences with other carriers. As illustrated by the line $S \propto A^{-1}$, the overall decrease of $S$ with $A$ follows the prediction (Eq. 16) quite well not only for the steady-state profiles, but also for the carriers obtained from the snapshot. As a main difference, non-steady river segments between confluences of carriers do not follow the steady-state relation $S \propto A^{-1.5}$ exactly. However, this difference apparently has no effect on the overall decline in steepness, $S \propto A^{-1}$, where

it even seems that the rivers from the snapshot are slightly closer to this relation than the steady-state carriers.

Actual and steady-state elevations are compared in Fig. 8. The data were again obtained from the 401 snapshots. In order to quantify the average behavior, mean elevations along lines in the $x_2$-direction are considered instead of individual data points.

If the average over the entire area is considered, so over all carriers and redistributors at a given $x_1$-value, the equilibrium elevations are much lower than the actual elevations. This finding is not surprising since almost the entire foreland area is covered by redistributors and their catchments. These are flat in equilibrium, where the catchments of redistributors draining directly into the ocean are even at zero elevation.

In turn, the mean elevation along the actual carriers is quite close to the respective equilibrium elevation. The maximum relative deviation occurs far away from the mountain range, where the mean elevation is about 80 % of the equilibrium elevation (dark blue domain). However, absolute elevations are small here, so that the deviation is also small on an absolute scale. The ratio approaches 1 toward the mountain range, where it is even between about 0.99 and 1.01 in the green, orange, and yellow regions. This result confirms the observation that the largest rivers are described reasonably well by the concept of carriers in equilibrium (Fig. 7).

## 7 Source-to-sink considerations

The deviation in elevation between the actual carriers and the respective equilibrium profiles seems to be unimportant at first. However, it implies that carriers are less steep than expected close to the ocean and thus deliver less sediment

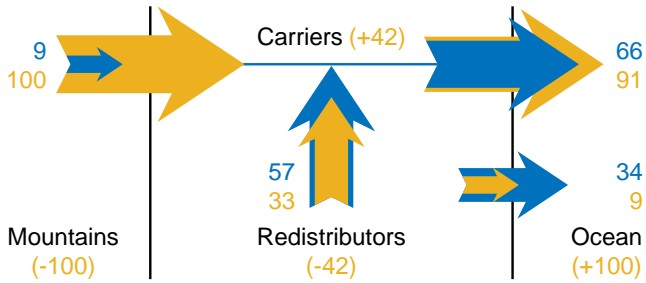

**Figure 9.** Balance of water and transported sediment. Blue arrows describe discharges, equivalent to catchment sizes. The values are normalized to the total catchment size and expressed in percent, so that 100 corresponds to the total catchment size. Orange-colored arrows describe sediment fluxes normalized to the sediment flux from the mountain range (also expressed in percent). The numbers in parentheses describe the sediment balances of the individual system components.

to the ocean in total than they receive from the mountain range. This deficit is related to the drainage pattern, where a considerable part of the foreland is directly drained into the ocean by redistributors. So a part of the sediment flux from
5 the mountain range is deposited by the carriers and transported into the ocean by redistributors after the network has reorganized.

Figure 9 shows the mean sediment budget obtained from the 401 snapshots, where all discharges (blue arrows, equiv-
10 alent to catchment sizes) are expressed as percentages of the overall discharge and all sediment fluxes (orange-colored arrows) as percentages of the sediment flux from the mountain range. One-third of the total domain (including the mountain range) is drained directly into the ocean by redistributors.
However, the contribution of these rivers to the total sediment delivery is only 9 % since their catchments are rather flat. So the carriers deliver more than 90 % of the sediment flux from the mountain range to the ocean.

However, this result does not imply that the carriers are
20 routing more than 90 % of their sediment flux to the ocean and deposit less than 10 % to be cleaned up later by redistributors. Here, the sediment flux from the redistributors into the carriers also plays an important part. This flux amounts to 33 % of the influx from the mountain range. So the to-
25 tal sediment input to the carriers is in fact 133 %. Therefore, the 91 % delivered to the ocean are less than 70 % of the total sediment input, while about 30 % (42 out of 133) are deposited.

So the simple concept of depositing sediment by the car-
30 riers and cleaning up the deposits later by the redistributors contains an important internal component. In sum, the redistributors erode as much material as deposited by the carriers on average, equivalent to 42 % of the sediment flux from the mountain range. Only a quite small fraction of this sediment
flux is delivered to the ocean (here about $\frac{9}{42} \approx 21\,\%$), while the majority arrives in the actual carriers. A considerable part

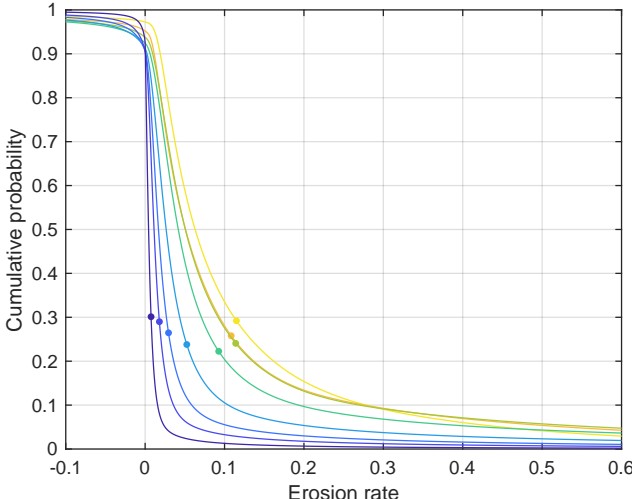

**Figure 10.** Cumulative distribution of the erosion rates of redistributors. The curves correspond to the domains defined in Fig. 1. The dots depict the respective mean net erosion rate.

of this material is deposited further downstream and waits there to be eroded by a new generation of redistributors after the network has reorganized.

It should, however, be emphasized that these numbers do
not only depend on the considered model, but also on the size of the foreland in relation to the size of the mountain range. In order to obtain a sufficiently large region far away from the mountain range, the foreland is in total ten times as large as the mountain range. For a smaller foreland re-
gion, the fluxes from the redistributors into the carriers and the respective balances ($\pm42\,\%$ of the sediment flux from the mountain range) would be smaller.

The quite large sediment flux from redistributors into carriers also explains the high rates of sediment deposition in
carriers, which will be investigated in the following section. As found in Sect. 6, longitudinal profiles of carriers are described well by equilibrium profiles on average. This means that they are able to transport almost the entire material eroded in the mountain range on average. Only in combi-
nation with the additional sediment supply from the redistributors, the total amount exceeds the transport capacity considerably, which enforces rapid deposition.

## 8  Rates of erosion and deposition

Figure 10 shows the cumulative distribution of the erosion
rates of all redistributors, where negative rates refer to deposition. These rates were evaluated in each step of the simulation from $t = 100$ to $500$ (so not only at the snapshots). So the rates are average rates over time intervals of $\delta t = 2^{-10} \approx 10^{-3}$ (typically some hundred years).
It is immediately recognized from the cumulative probabilities at zero erosion rate that the vast majority of all redis-

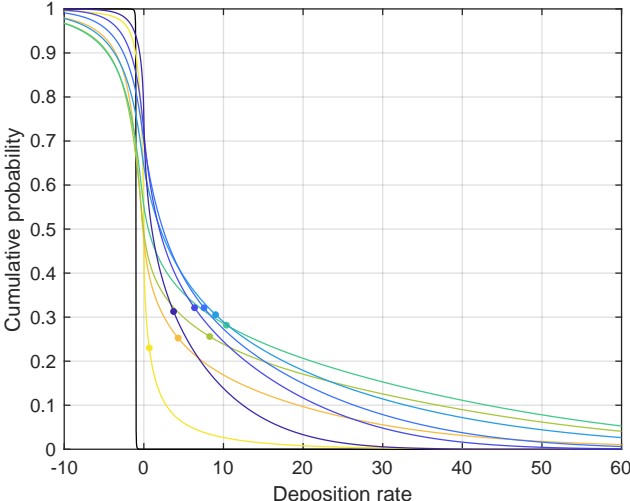

**Figure 11.** Cumulative distribution of the deposition rates of carriers. The curves correspond to the domains defined in Fig. 1. The dots depict the respective mean net deposition rate.

tributors is indeed eroding as discussed earlier. The fraction of eroding redistributor sites is greater than 90 % everywhere and even greater than 97 % close to the mountain range (yellow curve).

The rates are overall rather low compared to the mountain range ($E = U = 1$). In the four domains close to the mountain range (yellow to green curves), the net mean erosion rate (where deposition contributes negatively) is about 0.1, so about 10 % of the erosion rate in the mountain range. Further away from the mountain range (blue curves), the rates decrease rapidly. The average erosion rate over all redistributors is 0.042. Taking into account that the foreland area is ten times as large as the mountain range and covered almost entirely by redistributor sites, this result could also be obtained directly from the balance of the redistributors shown in Fig. 9 (total erosion amounting to 42 % of the sediment flux from the mountain range).

The respective rates for the carriers are shown in Fig. 11. As carriers predominantly deposit sediments, rates of deposition are shown here, where negative rates describe erosion. The rates are overall high compared to the redistributors and also compared to the mountain range. The overall mean rate is 4.8, so almost five times higher than the erosion rate in the mountain range. Since the carrier sites cover only a small part of the area, this result cannot be derived directly from the overall balance (Fig. 9).

In contrast to the redistributors, the lowest mean net rates occur close to the mountain range. The yellow domain is the only region where the mean net rate of deposition ($\approx 0.7$) is lower than the erosion rate in the mountain range. As shown in Fig. 4, the lateral mobility is also small in this region, at least for the largest river.

The rates increase rapidly with increasing distance from the mountain range and reach a mean net rate greater than 10 in the turquoise domain. This domain extends to a distance of 512 pixels from the mountain range, equal to the full width of the mountain range. Both deposition and erosion rates are quite high here. Almost 50 % of the carriers in this domain are depositing at rates higher than the erosion rate in the mountain range, and 30 % of them are eroding at such rates. The rate of deposition is even higher than 60 times the erosion rate in the mountain range at more than 5 % of the carrier sites here. With regard to these high rates of deposition, we must keep in mind that the model does not include floodplains explictly. Since sediments are deposited pixel by pixel, wider floodplains can only be filled by rapid avulsions. This may result in an overestimation of rates of deposition.

Further away from the mountain range, the rates decrease (blue domains). However, this decrease is much slower than for the redistributors, and the mean rates of deposition stay clearly above the erosion rate in the mountain range. The occurrence of a maximum in deposition rate at some distance from the mountain range is related to the drainage pattern. Close to the mountain range, almost all rivers are carriers. So a rather small sediment flux from the redistributors is distributed among a large number of carriers and thus has little effect compared to the sediment flux from the mountain range.

The strong autocyclicity in the sediment deposition by carriers raises the question whether it may even shadow effects of climatic variations. Recently, Yuan et al. (2022) investigated the effect of periodic oscillations in precipitation with a similar model and found a clear link between modeled climate oscillations and sediment signals. While a variation of some tens of meters in the mean elevation of the river bed was found at the edge of the mountain range, the respective rates of aggradation are rather low. Assuming a relative variation in precipitation by $\pm 50$ %, the maximum rate of aggradation is about $0.25 \ \mathrm{mmyr}^{-1}$ at an uplift rate of $1 \ \mathrm{mmyr}^{-1}$ (independent of the period, estimated from their Fig. 2). This rate is already lower than the mean rate found here for the very proximal region of the alluvial fans and by more than a decade lower than the mean rates in the more distal regions. So autocyclicity will probably dominate the record at short time scales and small spatial scales except for the very proximal region of the alluvial fans. However, finding out which scales are needed to overcome the dominance of autocyclic aggradation requires further investigations.

## 9   The time scale of network reorganization

While the interplay of carriers and redistributors is the key to understanding the long-term reorganization of the drainage network, the respective time scale cannot be determined directly from the rates of erosion and deposition. In this section, a simple concept is used for quantifying the reorgani-

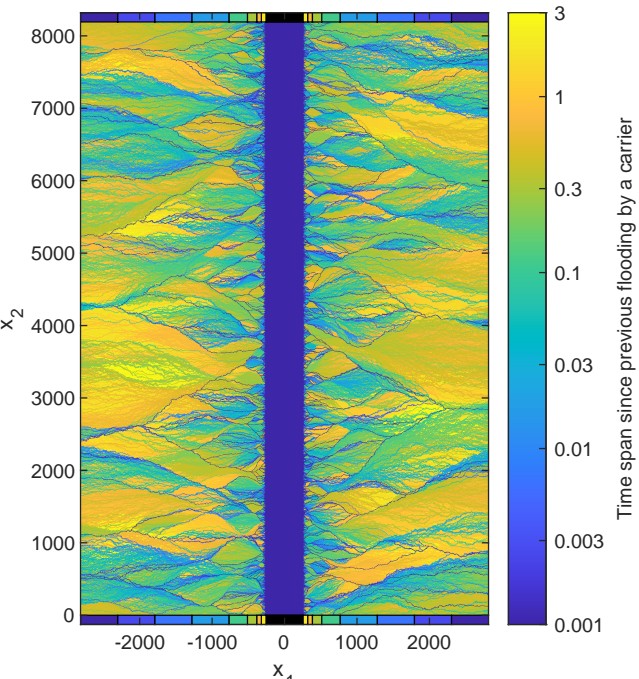

**Figure 12.** Time span since the previous flooding by a carrier for $t = 128$.

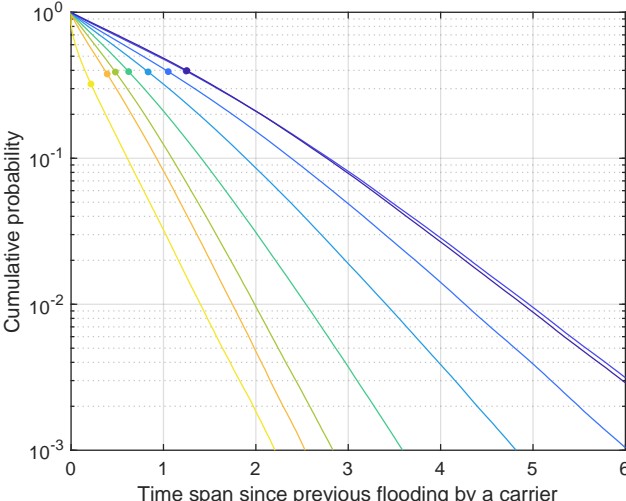

**Figure 13.** Cumulative distribution of the time span since the previous flooding by a carrier. The colors correspond to the regions defined in Fig. 1, and the dots depict the mean time span.

zation of the drainage network and its potential fingerprint in the deposits. Knowing that carriers are predominantly depositing sediment and redistributors are predominantly eroding, the time span since the previous flooding by a carrier is investigated. Figure 12 shows a map of this time span for the snapshot from Fig. 1. The spatial pattern is irregular. While dark blue areas depict the mountain range and the actual carriers, there are more or less continuous blueish areas depicting large regions that were flooded by carriers recently. In turn, there are also yellow areas indicating that some regions were not flooded by carriers for more than 3 time units. Some of these yellow areas are intersected by darker lines, which means that individual carriers crossed the area later without affecting the area as a whole.

The cumulative statistical distribution of the time span since the previous flooding by a carrier is shown in Fig. 13. The data were derived from the 401 snapshots. If flooding by carriers was a random process where the probability is independent of the time span since the previous event, the time spans would follow an exponential distribution (a straight line in the plot). The results suggest an exponential distribution at large time spans.

The time spans since the previous flooding by a carrier increase from the mountain range toward the ocean. While the mean time span is about 0.2 in the yellow domain, is increases to about 1.25 in the two outermost domains. So we can expect older deposits at the surface further away from the mountain range.

As it was found in Sect. 3 that the frequency of avulsions of large rivers depends strongly on the time increment $\delta t$ used in the simulation, the time scales of network reorganization also depend on $\delta t$. However, this dependence is weaker than expected from the frequency of avulsions. While reducing $\delta t$ from $2^{-10}$ to $2^{-13}$ increases the frequency of avulsions of large rivers by a factor of almost 3, the mean time spans since the previous flooding by a carrier decrease only by about 20 to 30 %. So a large fraction of the observed avulsions seem to be related to either back-and-forth oscillations in flow direction owing to the single-neighbor (D8) flow-routing scheme or to filling floodplains pixel by pixel. In particular, the increase in the time span since the previous flooding by a carrier toward the ocean appears to be robust.

The increase in the time span since the previous flooding by a carrier might be interpreted as a decreasing rate of river avulsion toward the ocean, similarly to the decreasing rates of deposition. However, we also have to take into account the spacing of the carriers here, which also increases toward the ocean as recognized Fig. 6. The increase in spacing is even stronger than the increase in the time span since the previous flooding. So the individual rivers do not become less active toward the ocean, and the longer time spans arise from a smaller number of carriers sweeping over the area.

The distributions shown in Fig. 13 deviate from exponential distributions at short time spans. The rapid decline of the distribution in the yellow domain indicates a clustering of events in the sense that there is an increased probability that the river returns to a location where it was recently. A random walk is the simplest process with this property, which could here be diverting the river randomly in the regime of the alluvial fans close to the mountain range. Far away from the mountain range, the behavior is opposite. Here, the probabil-

ity of flooding by a carrier increases if the previous flooding was long ago. This effect is presumably related to the topography. Since carriers are predominantly depositing sediment and redistributors are eroding, large channels that have not been carriers for a long time are rather flat and are thus favored candidates for becoming carriers in the future.

Using cosmogenic Ne isotopes, Sinclair et al. (2019) obtained a strong variation in the residence times of pebbles about 1000 km downstream of the source region located in the Rocky Mountains. It was found that some of the investigated clasts had even spent some million years in the river. In the same study, a conceptual model for the recycling of pebbles by tributaries was proposed. In principle, this concept already contains the idea of carriers and redistributors implicitly.

Concerning the order of magnitude, the time scale of network reorganization obtained in this study aligns well with the transit times obtained by Sinclair et al. (2019). However, we should keep in mind that the time span since the previous flooding by a carrier is not equivalent to transit times of sediment clasts. The generic form of the shared stream-power model (Eq. 1) predicts only a net rate, but not the exchange of particles between the river and an alluvial cover. Therefore, tracking sediment clasts in OpenLEM would not only require some technical effort, but also an extension of the shared stream-power model itself.

Carretier et al. (2016) already added a component for tracking clasts to the model CIDRE, which defines erosion and deposition rates explicitly. Simulating the distribution of clasts in a smaller region (equivalent to the yellow, orange, and green domains in this study), a variation of transit times over several decades was found (Carretier et al., 2020). The obtained distribution may even be heavy-tailed with a power-law tail, consistent with results obtained from a canyon in the Central Andes (Carretier et al., 2019). In turn, the exponential distribution of the time span since the previous flooding by a carrier suggests that the reorganization of the drainage network alone would not produce heavy-tailed distributions. So finding out whether transit-time distributions of sediment clasts are indeed heavy-tailed and, if so, explaining their origin requires further research.

## 10 The effect of consolidation

So far, a transport-limited model was used for the foreland region. However, we have seen in the previous section that a considerable part of the area may be covered by deposits older than some 100,000 years, where the question arises whether eroding such deposits is described well by a transport-limited model.

In order to investigate the effect of a finite erodibility $K_d$ in the erosive regime, an extension of the numerical scheme proposed by Hergarten (2020) was developed and implemented in OpenLEM. This extension switches between the shared stream-power model with a finite erodibility $K_d$ and the transport-limited end-member ($K_d \to \infty$) at each node. By integrating the decision into the scheme, the fully implicit character of the scheme can be preserved almost completely. Only the base level of each node (the elevation of the flow target) has to be adopted from the beginning of the respective time step, while the actual values of all other properties can be included in the decision.

As an extreme scenario, the same parameters as in the mountain range were assumed for the erosive regime in the foreland ($K_d = K_t = 1$). This would be an instantaneous consolidation of all deposits to a rock with the same properties as the bedrock in the mountain range. Although unrealistic, this extreme scenario is useful for investigating the effect of not fully transport-limited conditions in the erosive regime.

It was already recognized in Sect. 7 that the sediment fluxes from the redistributors into the carriers are very important for the high rates of sediment deposition in the carriers. The mean erosion rates of the redistributors indeed decrease strongly, where a more than fivefold decrease was found except for the two regions closest to the mountain range (yellow and orange). This may be surprising at first because the effective erodibility (Eq. 6) is reduced only by a factor of 2 (from 1 to 0.5). However, it was already shown in the context of knickpoint migration that disturbances propagate upstream at a velocity defined by $K_d$, which explains the big difference between $K_d \to \infty$ and $K_d = 1$ in the transient behavior.

The reduction in mean erosion rate equivalently reduces the sediment fluxes from the redistributors into the carriers and into the ocean. As shown in Fig. 14, this reduction strongly affects the sediment balance of the carriers. Now the total sediment input is only 107 % instead of 133 % (Fig. 9), while 98 % are delivered to the ocean instead of 91 %. So 92 % of the total sediment input are delivered to the ocean and only 8 % are deposited. The respective ratio was about 30 % in the transport-limited model (Fig. 9). So moving from the transport-limited model to the shared stream-power model with $K_d = K_t = 1$ also reduces the rates of sediment deposition in the carriers considerably, although the decrease is not as strong as in the erosion rates of the redistributors. In agreement with the results of the sediment balance, the decrease is by a factor of 3.5 to 4.8, except for the first and last domain (yellow and dark blue), where it is less than 3.

The increase in the rates of erosion and deposition also slows down the dynamics of network reorganization. While the distributions of the time since the previous flooding by a carrier was found to be qualitatively similar to the distribution shown in Fig. 13, the time scale is stretched. The mean time since the previous flooding by a carrier increases by a factor between 4.6 and 5.8 for the individual domains. These factors follow the decrease in erosion rates in the redistributors rather than the decrease in deposition rates in the carriers. This finding emphasizes the relevance of the erosion by

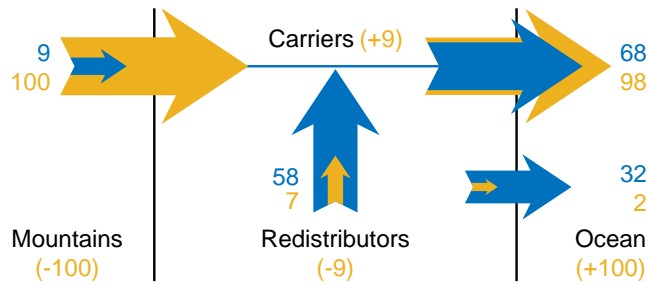

**Figure 14.** Balance of water and transported sediment for the scenario with instantaneous consolidation ($K_\mathrm{d} = K_\mathrm{t} = 1$ in the erosive regime). Blue arrows describe discharges, equivalent to catchment sizes. The values are normalized to the total catchment size and expressed in percent, so that 100 corresponds to the total catchment size. Orange-colored arrows describe sediment fluxes normalized to the sediment flux from the mountain range (also expressed in percent). The numbers in parentheses describe the sediment balances of the individual system components.

redistributors and the resulting sediment flux to the carriers for the dynamics of the foreland rivers.

## 11   Conclusions and outlook

This paper is intended to define some kind of reference scenario for fluvial landform evolution in a tectonically inactive foreland of a mountain range. Additionally, it can also be seen as a starting point for further studies.

The considered scenario combines one of the simplest models of large-scale fluvial erosion and sediment transport – the shared stream-power model – with a simple geometry consisting of a mountain range and an inactive foreland. In order to understand the behavior of the model, the foreland rivers were subdivided into two classes – carriers and redistributors. Carriers originate in the mountain range and are thus responsible for the large-scale sediment transport to the ocean. In turn, redistributors are rivers whose entire catchment is located in the foreland.

Using the concept of carriers and redistributors, it was shown that a steady-state topography in the strict sense is impossible in the foreland even under constant conditions. Although the topography becomes more or less constant on average over long times, the drainage network in the foreland permanently reorganizes. On the other hand, longitudinal profiles of carriers are described well by a hypothetical steady state on average, where the sediment flux from the mountain range is just routed to the ocean. The concavity index of carriers is typically greater than the concavity index $\theta = \frac{m}{n}$ of rivers in a mountain range at uniform erosion, but smaller than the value $\theta = \frac{m+1}{n}$ expected for carriers without sediment supply by tributaries. By analyzing the topology of the drainage network, it was found that $\theta$ is in the middle between these two values, so $\theta \approx 1$ for the linear version of the shared stream-power model ($n = 1$).

It was found that redistributors are predominantly eroding at rates lower than the erosion rate in the mountain range. In turn, carriers are predominantly depositing sediments, where the rates are typically much higher than the erosion rate in the mountain range. As a major result, the sediment flux from the redistributors into the carriers plays a central part for the deposition of sediments and for the reorganization of the drainage network. While the erosion rates of the redistributors are rather low, the respective areas are large, generating a considerable sediment input in total. As a consequence, the assumptions on the erosion in the foreland are more important for the dynamics of the rivers than it may seem at first.

While these results might be fundamental, there are two major limitations, which are related to each other. First, there is the finite length of the time increment used in the numerical simulation. Concerning the accuracy, much smaller time increments than used in this study would be desirable. This would, however, not only increase the numerical effort, but would also shift the time scale into a range where individual flood events become relevant in reality. As a second point, floodplains are not explicitly taken into account. Valleys are filled with sediments pixel by pixel in combination with a high frequency of avulsions. Finding an approach to extend deposition laterally over floodplains would probably also help to solve the issue with the time increment.

There are several further aspects where subsequent studies should go deeper. This also includes the consideration of transient states and the comparison to real-world topographies. In addition, rates of sediment deposition were only investigated at a given time scale. Given the numerical efficiency of the model OpenLEM used here, the scaling properties could be investigated over a range from some 100 years up to millions of years. As a next step, tracking ages of deposits in different depths would be interesting and would allow for a validation by real-world data, although technically more challenging.

In addition, the nonlinear version of the shared stream-power model (so with exponents $n > 1$) should also be investigated in subsequent studies. From a theoretical point of view, the still existing uncertainty concerning the exponent $n$ in the stream-power formulation is still a challenge. The theoretical considerations of the concavity index of carriers suggest that the exponent $n$ has an effect in the foreland even under spatially uniform conditions, in contrast to active mountain ranges. So a more thorough investigation of the influence of the exponent in combination with real-world river profiles may considerably contribute to our knowledge on the value of this exponent.

**Code availability.** All codes are available in a Zenodo repository at https://doi.org/10.5281/zenodo.6575654 (Hergarten, 2022b). Users who are interested in using the landform evolution model OpenLEM in their own research are advised to download the most recent version from http://hergarten.at/openlem (Hergarten, 2022a).

The author is happy to assist interested readers in reproducing the results and performing subsequent research.

**Author contributions.** N/A

**Competing interests.** The author declares that there is no conflict of interest.

**Acknowledgements.** The author would like to thank Jean Braun and two anonymous reviewers for their constructive comments and Sagy Cohen for the editorial handling.

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
