# Peer review of "Theoretical and numerical considerations of rivers in a tectonically inactive foreland"

_Earth Surface Dynamics, 2022_

## Author Response (AR1)

Dear Reviewers, dear Editor,

thank you very much for your constructive and encouraging comments! I am quite happy that all reviewers found the concept of carriers and redistributors useful. The points addressed in the three reports are discussed below, where changes to the manuscript are highlighted in bold letters. Line numbers refer to the version with highlighted changes.

In addition, I expanded the discussion about the relation between equilibrium sediment flux and catchment size a bit since it may be useful for other modeling approaches (lines 345–359).

Best regards,

Stefan

**Reviewer 1**

*General comments*

*The manuscript entitled Theoretical and numerical considerations of rivers in a tectonically inactive foreland is a timely contribution to the discussion of how fluvial sediment transport dynamics influence the transfer of sediment through landscapes and the implications for interpreting sedimentary records. Hergarten usefully identifies two types of rivers contributing to erosion and sediment transport within an alluvial foreland and presents a clear methodology for analysing the net contribution of these respective river types to the overall sediment budget and network morphology. It is the detailed explanation of the modelling approach that is a great asset to this paper and a reason for why I believe this paper is well suited for the Esurf community. While the paper does not integrate any field or lab data to support its modelling, I appreciate the adaption of the model to integrate typical field observations, such as the changing erodibility of sediment surfaces as they increase in age. I think that the insights gained from this modelling approach are a useful baseline for future studies.*

*While the paper is generally well written and easy to follow, there are a few sentences that would benefit from rewording, of which some are listed below in the technical comments. The grammar and some of the sentence structure could be improved by editing from a native English speaker. The figures are excellent and easy to interpret and the mathematical formulae are correctly defined. The first half of the abstract would be improved by stating more explicitly the context of the paper, as at present it is a little vague and difficult to follow. There are a few recent references that could also be included to give credit to parallel approaches being developed in this field (Malatesta et al., 2017 Basin Research, plus those detailed below).*

**I tried to improve the first sentences of the abstract (lines 1–4) and added some references to the introduction (lines 22, 31–32).**

*Specific comments*

*In the introduction, it would be useful to explicitly define what you mean by steady state in the context of this work and why this definition is relevant.*

**I tried to point out this aspect more clearly at the end of the introduction (lines 66–72).**

*The expectation of high concavity values ($> 1$) for these alluvial rivers was surprising for me, as I am not aware of these values being frequently observed in natural alluvial rivers (c.f. Wickert and Schildgen 2019 ESurf). Why is this model formulation acceptable for this case?*

I am even not aware of any serious data on the concavity of alluvial rivers without uplift and subsidence. Otherwise I would have used these. However, the concavity values predicted by my theory are typically still slightly $< 1$. Admittedly, I am not fully convinced by the modeling results of Wickert and Schildgen (2019). The problem with all these studies starting from physical relations is that we end up at a point where we need to calibrate exponents in order to arrive at a concavity index consistent with observations. In that study, it is after Eq. 51. There the conjecture is made that a transport-limited river without uplift and subsidence has the same concavity as a detachment-limited river at uniform uplift. This conjecture is not well in line with the "mainstream" (Whipple and Tucker 2002, Davy and Lague, 2009, Yuan et al. 2019, ...), where the old findings of Hack (1957) are rather interpreted in the way that detachment-limited and transport-limited rivers have the same concavity under uniform uplift. The conjecture made by Wickert and Schildgen (2019) shifts the concavity index towards smaller values and even makes transport-limited rivers straight under uniform uplift. Personally, I do not agree to this conjecture, but I do not want to criticize without being able to provide solid real-world data.

*I think the absence or distribution of accommodation space generation in the model needs to be specified and considered. The low deposition rates along the carrier rivers are perhaps to be expected if there is no accommodation space available for the sediment to be deposited in. I am wondering how the net fluxes and the interactions between the carrier and redistributing channels would shift as an accommodation space is defined. This maybe something to mention in the future work section of the conclusions as it has important implications for understanding the generation of physical and measurable parameters such as downstream grain size fining trends along alluvial rivers (for example, see Harries et al., 2019 ESPL). For the reference case in this paper, I would expect that any size-selective grain size fining that would occur along the length of the carrier rivers would be solely introduced by the integration or recycling of older sediments by redistributing channels or the migration of carrier channels.*

There is indeed no distinct accommodation space in the model, which is one of the major limitations. The implicit scheme used in this study automatically generates some accommodation space if the elevation of the trunk stream exceeds that of its tributaries. Then sediment is pushed back into the tributaries, but this effect is much weaker than I would expect for a flooded plain. However, the deposition rates are quite high along the carriers and would decrease if additional accommodation space was provided. **I added a few sentences on this aspect (lines 158–165, 186–188, 464–468, 599–602).** Anyway, a good perspective for future studies.

Recent work has demonstrated that sediment transport along alluvial rivers is a non-linear process, even at millennial timescales (Carretier et al., 2019 Sci Rep, Sinclair et al., 2019 Geology). In which case, net sediment transfer is not well characterised by mean values. It might be worth mentioning this work in the introduction and highlighting the usefulness of the reduced complexity approach presented. The observation that the storage time of sediment in the fan surfaces is highly variable and dependant on the distance between channels (section 9) aligns well with the findings of Carretier et al. (2020 EPSL).

To be honest, I do not understand why nonlinear sediment transport along alluvial rivers cannot be described well by mean values. In particular, mean sediment fluxes are still well-defined. **Anyway, I added a few sentences about these results to Sect. 9 since I found them very interesting (lines 522–539).**

Technical comments

Frequently, the phrase 'on the mean' or 'in the mean' is used (e.g. L289, L319 and L426). I think the desired phrase is 'on average.'

**Fixed, thanks (lines 284–285, 396, 432, 582, 584)!**

L21: 'modelling studies mentioned above' - include more references to modelling studies

**I added some more references and tried to bring some structure into the list (lines 30–33).**

L23-24: Reword

**I tried (lines 34–36).**

L55: by 'tectonically inactive' do you mean without subsidence?

Yes, also without subsidence. **I clarified it two sentences later (lines 76–78).**

L56 and L419: 'exposed' is perhaps not the right word for this context. You impose boundary conditions.

**I reworded it at both occurrences (lines 76–78, 577).**

L214: Remove 'in'

**Fixed (line 304), thanks!**

L289: Reword

**I tried (lines 396–399).**

L339: 'one half to one widths' - reword

**I tried (liness 458–459).**

L342: Replace 'rate' with 'range'

**Fixed (line 461), thanks!**

L429: Reword

**I tried and also tried to go a bit more into detail (lines 586–589).**

L440: Reword

**I tried (lines 606–608).**

**Reviewer 2**

*This manuscript presents a numerical model and derives a conceptual framework to quantitatively explore sediment transport dynamics and drainage network evolution over large spatial and temporal scales in a foreland setting.*

*An emerging concept of the study is the division of foreland rivers into carriers and redistributors with well-defined geomorphic functions, long profiles, and sediment transport dynamics. The identification of the two groups of rivers is appealing and has the potential to serve other studies that explore sediment routing in foreland basins.*

*The manuscript presents only numerical results. This is a valid choice  numerical insights could be highly beneficial even when presented independently from a field, experimental, or previously presented numerical research questions or observation. However, I believe that the current manuscript could greatly benefit from some connection(s) to field observations, illustrating that the new concept of carriers and redistributors is meaningful.*

*Aside from a few odd phrasing choices (pointed out in RC1), the manuscript is overall well-written.*

*There are two major issues that I recommend considering, which could potentially increase the manuscripts impact and usefulness.*

*First, the model is not described in sufficient detail. While the model was presented in a previous study, readers should be able to get the main message of how the model works without needing to consult the previous manuscript. This is particularly true since much of the inferred sediment and drainage dynamics appear to be specific to the model. Equation 1 has two unknowns: E and Q, and additional equations are needed to close the system. Both unknowns can be formulated as functions of local elevation, but this is not currently specified. Boundary conditions are also not fully specified. I.e., how does the model deal with Q at the highest node of each network? Choices pertaining to drainage dynamics are not presented and discussed. I.e., how the model deals with local slope reversals due to sediment deposition? Does the model assume steepest descent to induce drainage change? What is assumed about flow routing? Are lakes allowed? These choices likely directly influence model results, but readers are currently left in the dark.*

Originally, I thought of mentioning only those aspects that are really specific to the model and important for understanding the meaning of the parameters. However, since two reviewers suggested some extensions, **I added a few lines on closing the system, internal boundaries, and flow routing (lines 91–101) and about lakes and reverse sediment fluxes (lines 156–165).**

*The model is developed with non-dimensional quantities, and a specific dimensional interpretation is proposed. This is a common practice that works in many cases. However, in the current manuscript, I found that the repeated dual interpretation (dimensionless and dimensional) is confusing. One way to overcome this issue is to formally present the scale factors once (it could also be interesting to see a non-dimensional analysis of equation 1) and from that point on to present the results only with either dimensional or non-dimensional form (the former is probably easier to read). On the same issue: why does it make sense to use two length scales? Why can't the analysis work with a single length scale?*

I thought that the dual consideration would make it easier for the readers, but I accept that two reviewers do not like it. Completely switching to dimensional values is not an option for me since it is an unnecessary restriction. **So I stated the scaling factors more clearly (lines 166–188) and use only nondimensional values except for a few occasions where it is important to imagine a real-world order of magnitude.** And of course it makes sense to allow horizontal scaling and vertical scaling independently (as long as the model allows to do so). Otherwise, rescaling the topography to a given maximum elevation (or something similar) would also define the horizontal extension of the domain, which would impose an unnecessary constraint.

*Some of the steady-state and long profile analyses presented in section 6 could be moved to the model description, providing much-needed intuition of model expected behavior.*

Might be helpful, but I am afraid that this would be a mess.

*The second major issue is that I struggled to balance sediment mass and topography across sections 7-9. I assume that the model conserves mass (including sediments deposited in the ocean). However, I could not balance it myself (I didn't try to balance it formally, but just in terms of sources, sinks, processes, and orders of magnitude). Figures 9 and 14 are confusing to me. How come sediments are incorporated from the foreland, but there is no arrow showing sediments being damped in the foreland? How come the percent sum of sediments deposited in the ocean is 100 while both the figure and the text refer to the total sediments as more than 100% (e.g., the 91% of the carriers is out of the 133% coming from both the mountain and the foreland).*

As two out of three reviewers struggled with the sediment balances in Figs. 9 and 14, I must accept that it is not as trivial as I thought. **So I labelled the carriers and redistributors in the figures and added the net balance of each domain. I hope that it is clear now in combination with some additional text (lines 414–415, 446–448).** In short: Redistributors erode 42 from the foreland and transfer 33 to the carriers and 9 to the ocean ($42 = 33 + 9$). Carriers receive 100 from the mountains and 33 from the redistributors, of which 91 are given to the ocean and 42 dumped in the foreland ($100+33 = 91+42$).

*Similarly, I'm unsure how to reconcile figures 10, 11, and the sediments transported to the ocean. Carriers' deposition rate is larger by an order of magnitude with respect to the erosion rate of the mountain and by two orders of magnitude with respect to the erosion rate of redistributors. How can that be? Are the carriers two orders of magnitude smaller in area than redistributors? How is it possible that with such a high sedimentation rate, the foreland is not growing in topography but reaches a statistical steady state? How can this be reconciled with figure 9?*

It is exactly the way you wrote it. The area covered by the carriers is small, while the redistributors fill almost the entire foreland region. All redistributors together erode an amount of 42 (in percent of the total sediment flux from the mountains) from the foreland, while all carriers together deposit the same amount. And of course, this is only compatible with a statistical steady state only if carriers and redistributors switch their roles from time to time. **I added a note in order to clarify the relation between the rates and the sediment balance (lines 446–448, 452).**

*Perhaps a formal statement (with equations) of mass/volume conservation could help out here, clarifying which component out of the mass balance each figure analyzes?*

Rather not – hiding behind equations that would need additional symbols cannot be the solution here. Since it is not complicated, it has to be explained in words.

*Line comments*

*Line 1 – First sentence of the abstract reads a bit detached.*

**I added two sentences about the context (lines 1–4),** but I still prefer to start from the modeling perspective.

*Line 2 - 'Fundamental properties'. At this stage, only 'fundamental numerical properties'.*

For me, the numerical simulation is the way to elucidate the properties, but the properties are not numerical. **I added "theoretically and numerically" at the end of the sentence (line 6).**

*Line 100 - Not sure there is a need to mention simulations that are not presented here.*

Not necessary, but in my opinion useful for readers working in the same direction. Being aware of this issue before starting simulations taking several weeks would have saved time.

*Line 121 - Perhaps 'representation' instead of 'coordinates'.*

**Ok (line 166)**, although I was happy with "coordinates" in my previous papers.

*Perhaps a more informative title to section 2.*

**Ok, perhaps "model setup" (line 74)**, although it is probably not much better than "approach".

*Figure 3 - Maybe variance would be a better description than relief.*

Initially, I indeed used the standard deviation and also tested the ratio of standard deviation and relief. However, I finally found relief more intuitive and better for the situation of deeply incised valleys in the mountain range.

*Figure 5 - Maybe also show some profiles?*

In an early phase of writing, I plotted some raw profiles from the snapshots (Fig. 1) and from the hypothetic steady state (Fig. 5). However, the profiles look just like river profiles and are not very exciting. The respective slope-vs-area profiles of the rivers marked by the dots are already plotted in Fig. 7.

**Reviewer 3 (Jean Braun)**

*I have read your manuscript with great interest. It contains some very interesting results. In particular, I appreciated your division of channels in the foreland into carriers and redistributers. These are indeed very useful concepts to understand most of the dynamics of the quasi-steady sedimentary system you study and that you labeled a "tectonically inactive foreland". The analysis of the concavity of channels is very interesting and novel, the dual role played by the redistributers as well as the source-to-sink description of the system are all facilitated by the new nomenclature.*

*Although I support the publication of this material in ESURF, I express below some concerns that I have about the presentation of your results, their robustness and their applicability.*

*The objectives of the manuscript are relatively well explained and certainly interesting for those of us that like to play with equations. I find, however, that you could improve your introduction by relating better your objectives (last paragraph of the introduction) to questions that are asked by sedimentologists, geomorphologists. Similarly, I believe your paper would gain much in its impact if you were to come back to these questions (and how your work has contributed to their resolution) in your discussion.*

**I tried to improve it (lines 56–73, 474–483, 522–539)**, but I am not good in this field.

*Although the basic evolution equation has been presented elsewhere, I believe it is important for the comprehension and the flow of the manuscript to at least present the basic PDE that you are solving. Even though you focus on the quasi steady-state solution, you must solve an evolution equation, most likely expressed in terms of the vertical elevation of the topography as the main unknown. It is also important to give the form of this equation in the case where Kd tends to infinity, because this is the form that you have used for most of the results presented here (in the basin). Am I right in assuming that it then takes the form of a non-linear diffusion equation? For both the mathematically-oriented readers of your manuscript and the sedimentologist who might be interested in interpreting your results, it seems important to me that these equations (the full form and its asymptotic form when Kd tends to infinity) be presented.*

Right, although the numerical treatment is even fully time-dependent and does not focus on quasi-steady states. **I recapitulated the equations required for a closed system (lines 91–101) and added some information on the transition to the transport-limited end-member (lines 110–111).** However, I prefer not to repeat the equations explicitly for the limit $K_t \to \infty$.

*Although I fully support the need to use dimensionless variables when presenting model results, I do not agree with your approach to quote absolute values for basic parameters such as K or grid size and derive other length scales and time (or rate) scales out of it. I believe it would be much more useful to explain with some simple relationships how the dimensionalisation should be done, i.e., how one could apply your dimensionless results to a problem of known size and rate.*

I thought that the dual consideration would make it easier for the readers, but I accept that two reviewers do not like it. **So I stated the scaling factors more clearly (lines 166–188) and use only nondimensional values except for a few occasions where it is important to imagine a real-world order of magnitude.**

*In the model description, you mention that the algorithm you use is implicit and thus unconditionally stable. You do not, however, assess its accuracy, which we know must depend on the time stepping (and grid spacing). Can you please provide us with an estimate of this accuracy. I am concerned (see point 6 below) that the solution might be dependent on the step size. If your results are applicable to natural systems, which are characterized by finite avulsion rates, the model should be characterized by a characteristic time for channel geometry to change. I believe that you need to check whether the time step you are using is smaller than such a characteristic time for many of the conclusions you draw to be correct.*

I performed a test with $\delta t = 2^{-13}$ (8 times smaller than the original $\delta t$. As expected, the frequency of avulsions is strongly affected by $\delta t$, in particular for large rivers. **I added the respective numbers and some warning remarks (lines 195–204, 212–220, 502–508, 596–599).** However, it seems that the strong dependence on $\delta t$ mainly reflects the filling of floodplains by rapid avulsions and has a weaker effect on the large-scale dynamics (see also the point after the next point).

*You note that the foreland is made of two parts (a fan and what I will call an alluvial plain connecting the fan to the ocean). You also note that the behavior of the system is rather contrasted in these two sections. So what controls the size of the fan becomes an important factor in describing the system's behavior. I recently demonstrated with a 1D version of a model identical to yours that it is the size of the mountain catchment area that controls the size of the fan (regardless of the value of Kf). This implies that the setup you have used (with a very small mountain) leads to a relatively peculiar situation that might not be representative of many forelands. May I suggest that you test the robustness of your finding against the size of the fan (by changing the size of the mountain area). It might lead to very similar results with a simple shift of some of your curves (as shown in Figure 10). But it might not. Furthermore, some of the numbers you quote in your "source-to-sink" section may be quite different for a different relative size of the fan.*

I think we must discuss the sizes of the fans for future work since we probably obtain similar results for different reasons. In my results, I see qualitatively that the spacing of the biggest rivers leaving the mountain range defines the sizes of the fans in my simulation. Along the fans, the big rivers just capture smaller rivers. The region dominated by fans ends where big carriers of similar sizes join. Of course, the spacing of the biggest rivers is related to their catchment size. You mention that you demonstrated that the size of the mountain catchment controls the size of the fan. In your ESurf paper, however, it rather seems to me as if you enforce this result by a specific assumption on the catchment size. If I read it correctly, you apply Hack's law to the part of the rivers outside the mountain range alone and then simply add the catchment size $A_0$ of the part located in the mountain range (your Eq. 10, $A = A_0 + kx^p$). If we did this for any point in a "regular" catchment, it would be wrong. A short analysis of the biggest river from my simulation (5 snapshots, Figs. 1 and 2) does not confirm your hypothesis of a new Hack's law starting at the edge of the mountain range. While we indeed see a smaller increase of $A$ with $L$ close to the mountain range in the left-hand diagram (right of the dots), $A - A_0$ vs. $L - L_0$ (right-hand plot) differs strongly from your conjecture. **Add added a few sentences about the fan sizes (lines 262–270),** but the question who is right and whether it makes a difference will remain open. Please let me know whether my wording "by assuming a specific relation between catchment size and river length in the foreland" is ok for you. **In addition, I added a remark on the dependence of the balances on the size of the foreland region (lines 419–429).**

[Figure]

Figure 1: Test of Hack's law at 5 snapshots of the largest river from the mountain range. Left: full river profile. Dots refer to the edge of the mountain range. Right: starting from the edge of the mountain range as assumed by Braun (2022).

*Let me now come to my main concern: I found the part concerning the time scale for drainage reorganization very interesting. However, I do not know how to interpret these results to understand how real (natural) systems behave. I am particularly concerned about how the spatial and temporal resolutions of your model experiments might influence your results. I think this needs to be investigated for your results to have the impact they deserve. As channels have no width, there is a possibility that you might not be able to extract an avulsion time scale out of the basic equations, in which case many of the results you present (for the time evolution of the system in its quasi steady-state) might be difficult to use to interpret natural systems.*

I was surprised to see that the effect of $\delta t$ on the results of Sect. 9 is by far not as big as we would expect from the rates of avulsion (Fig. 3). While the frequency of avulsion increases by almost a factor of 3 if $\delta t$ is reduced from $2^{-10}$ to $2^{-13}$, the times analyzed in Fig. 13 only decrease by about 20 % (close to the mountain range) to 30 % (far away from the mountain range). This is much less than the effect of consolidation discussed in Sect. 10. So it seems that the short-term filling of floodplains by rapid avulsions strongly depends on $\delta t$, while the long-term and large-scale evolution does not. **I added some remarks on this aspect (lines 502–508),** being aware that this aspect requires further research.

*Another point of concern is your use of a single direction flow routing algorithm, which you should try to be better justify in your method description in a low slope system/environment controlled by continuously evolving states of deposition and erosion. Such natural systems are often characterized by non-dentritic channel networks with flow splitting occurring as often as flow merging.*

There are indeed some rapid back-and-forth avulsions in the model, which means that the model tries to mimic flow splitting occasionally. The question whether it makes a difference to "true" flow splitting is, of course not trivial. I think that the effect only becomes important in delta regions, but of course I cannot provide a serious justification for neglecting flow splitting. **I added two remarks on this aspect (lines 355–358, 506–508).**

*I also have some minor comments on the presentation of your results:*

[Figure]

Figure 2: Figure 13 for different $\delta t$.

Line 182: I am not sure about the use of the term "black and white scenario" nor to what it corresponds to.

Probably not a good term here. **So I removed it (line 272).**

Figure 9 is not clear; you have two sets of arrows leaving the foreland into the ocean; I believe one must be associated to carriers and the other to redistributors. This should be indicated somewhere (in the caption?). Also I am not sure to what corresponds the set of vertical arrows in the foreland? In a steady-state solution, on average the flux in and out of the foreland must be nil.

As two out of three reviewers struggled with the sediment balances in Figs. 9 and 14, I must accept that it is not as trivial as I thought. **So I labelled the carriers and redistributors in the figures and added the net balance of each domain. I hope that it is clear now in combination with some additional text (lines 414–415, 446–448).** The carriers deposit an amount of 42 in total, and the redistributors erode the same amount.

Please make sure that figure 13 that uses the classification of "regions" in the foreland as described/shown in figure 1 has a reference to it in the caption.

**Fixed, thanks!**

---

## Author Response (AR2)

Dear Sagy, dear Copernicus team,

thank you very much for your positive response! I did not find many grammatic issues, and the only native speaker in our institute would probably go mad if he was asked to read all papers of all groups completely. Anyway, my experience with the copy-editing team has been very good, so that they will surely find the lingering issues.

As suggested by Polina Shvedko, I changed the folor scheme of Fig. 7 and the respective dots in Figs. 1 and 5. However, the colors are not very relevant in this figure and none of the interpretation relies on the colors. In addition, I replaced red by orange in Figs. 9 and 14 since it seems to be better in monochromatic view.

Best regards,

Stefan